# Early detection of chronic hepatitis B and risk factor assessment in Turkish migrants, Middle Limburg, Belgium

Özgür M. Koc[1,2,3]*, Cécile Kremer[4], Niel Hens[4,5], Rob Bielen[1,2], Dana Busschots[1,2], Pierre Van Damme[6], Geert Robaeys[1,2,7]

1 Department of Gastroenterology and Hepatology, Ziekenhuis Oost-Limburg, Genk, Belgium, 2 Faculty of Medicine and Life Sciences, Hasselt University, Hasselt, Belgium, 3 Department of Medical Microbiology, School of Nutrition and Translational Research in Metabolism, Maastricht University Medical Centre, Maastricht, The Netherlands, 4 Interuniversity Institute for Biostatistics and Statistical Bioinformatics (I-Biostat), Data Science Institute, Hasselt University, Hasselt, Belgium, 5 Centre for Health Economic Research and Modelling Infectious Diseases, Vaccine and Infectious Disease Institute, University of Antwerp, Antwerp, Belgium, 6 Vaccine & Infectious Disease Institute, Centre for the Evaluation of Vaccination, Antwerp University, Wilrijk, Antwerp, Belgium, 7 Department of Gastroenterology and Hepatology, University Hospitals KULeuven, Leuven, Belgium

* ozgur.koc@uhasselt.be

**Data Availability Statement:** Data are available from the following repository: Koc, ÖM et al. (2020), Early detection of chronic hepatitis B and risk factor assessment in Turkish migrants, Middle

## Abstract

### Background

Turkey is an intermediate hepatitis B virus (HBV) endemic country. However, prevalence among Turkish migrants in Belgium is unknown, especially in those born in Belgium with a foreign-born parent, i.e. second-generation migrants (SGM).

### Aims

To evaluate the prevalence of HBV infection and associated risk factors in Turkish first-generation migrants (FGM), i.e. foreign-born, and SGM.

### Methods

Between September 2017 and May 2019, free outreach testing for hepatitis B surface antigen (HBsAg), hepatitis B core antibodies (anti-HBc), and antibodies against HBsAg was offered to Turkish migrants in Middle-Limburg, Belgium. Face-to-face questionnaire assessed HBV risk factors. HBsAg positive patients were referred and followed up. Turkish SGM were stratified into birth cohort born before and after 1987, since those born after 1987 should be covered by the universal infant vaccination program.

### Results

A total of 1,081/1,113 (97.1%) Turkish did go for HBV testing. Twenty-six (2.4%) were HBsAg positive; 11/26 were unaware of their status and 10/11 were successfully referred. HBsAg prevalence was 3.0% in FGM and 1.5% in SGM, $p$ = .070. Only one out of seven HBsAg positive SGM was born after 1987. In the multiple generalized estimating equations

Limburg, Belgium., Dryad, Dataset, https://doi.org/
10.5061/dryad.4f4qrfj87.

**Funding:** This study was sponsored by Gilead
Sciences. The funders had no role in the study
design, data collection and analysis, decision to
publish, or preparation of the manuscript.

**Competing interests:** ÖK received travel grants
from Gilead Sciences and his institution received
grants from Gilead Sciences, AbbVie, MSD and
CyTuVax B.V. NH is holder of the chair in evidence-
based vaccinology supported though a gift by
Pfizer. All outside the submitted work. RB has
received travel grants from AbbVie, Gilead
Sciences and MSD to attend scientific congresses
and research grants from Gilead and MSD. DB has
received travel grants from AbbVie and a research
grant from Gilead. PVD acts as chief and principal
investigator for vaccine trials conducted on behalf
of the University of Antwerp, for which the
University obtains research grants from vaccine
manufacturers; speakers fees for presentations on
vaccines are paid directly to an educational fund
held by the University of Antwerp. PVD receives no
personal remuneration for this work. GR has
received research grants from AbbVie, MSD,
Janssen Pharmaceuticals, and has acted as a
consultant/advisor for AbbVie, MSD, Gilead
Sciences and Bristol-Myers Squibb. This does not
alter our adherence to PLOS ONE policies on
sharing data and materials.

model, the most important risk factors for anti-HBc positivity were male gender ($p = .021$), older age ($p < .001$), FGM ($p < .001$), low educational level of the mother ($p = .003$), HBV infected mother ($p = .008$), HBV infected siblings ($p = .002$), HBV infected other family member ($p = .004$), gynaecological examination in Turkey or unsafe male circumcision ($p = .032$) and dental treatment in Turkey ($p = .049$).

## Conclusion

Outreach testing was well-accepted and referral to specialist care was generally successful. National HBV screening should be implemented in the Turkish FGM population and might be considered in SGM not covered by primary prevention strategies.

## Introduction

Hepatitis B virus (HBV) infection is one of the most common infectious diseases worldwide. Approximately one third of the world population has been exposed to the virus and an esti-mated 257 million people are chronically infected [1]. Regardless of the asymptomatic nature of chronic HBV infection, patients remain infectious to others and are at risk of death from cirrhosis and hepatocellular carcinoma [1]. In 2015, HBV-related complications accounted for 887,000 deaths globally [1].

The possibilities for antiviral treatment of chronic HBV infection have greatly improved over the past decades, and cost-effective drug therapies are now available [2, 3]. However, due to the asymptomatic course, most patients are unaware of their HBV infection and do not benefit from treatment. Screening for hepatitis B, aimed at early detection of chronic HBV infection with follow-up and antiviral treatment of eligible patients, is needed to reduce the economic burden of HBV-related complications and health-related suffering in Europe [4, 5]. In addition, screening could identify persons who are susceptible to infection and would bene-fit from vaccination [5].

In low endemic countries (< 2% hepatitis B surface antigen (HBsAg) positive), migrants born in intermediate (2–7.99% HBsAg positive) or high endemic (≥ 8% HBsAg positive) regions are an important risk group for chronic HBV infection [6]. In Belgium, the prevalence of chronic HBV infection is estimated to be 0.67% in the general population and 4.69% in first-generation migrants (FGM) born in intermediate or high endemic countries [6–8]. Migrants from intermediate or high endemic countries account for more than half of the patients with chronic HBV infection in Belgium [9]. Turkish individuals form the second larg-est FGM population in Europe and with a HBsAg prevalence in Turkey ranging from 2–3% in the Western part up to 7–8% in Eastern part, a substantial proportion of the Turkish migrant population may be infected with HBV [9, 10].

Most studies concerning hepatitis B testing in FGM involved southeast Asians in the USA, Canada and Australia [11]. A cross-sectional study in Turkish FGM in Germany showed a prevalence of 5% for chronic HBV infection, and this was 3% in a study conducted in the Netherlands [12, 13]. However, HBV risk factors and follow-up in Turkish FGM are mostly unclear. Furthermore, much less is known about infection prevalence, risk factors and follow-up in second-generation migrants (SGM), i.e. born in Belgium with a foreign-born parent. This is all the more important as the descendants of the FGM population grow. Regarding acute hepatitis B notification rates, a Dutch study showed a higher rate of 3.7/100,000 in SGM compared with 1.6/100,000 in native Dutch/Western persons [14].

We organized a hepatitis B testing and risk factor assessment campaign for the Turkish population in Middle Limburg, Belgium, a region with a relatively large Turkish population [15]. Here we report the prevalence of HBV infection in individuals with a Turkish background (FGM and SGM), the results of a questionnaire on risk factors, and the follow-up of patients referred to care.

## Patients and methods

From 1 September 2017 until 2 May 2019, we recruited ≥ 18 year-old Turkish migrants living in the region Middle Limburg. Follow-up was concluded on 2 November 2019, 6 months after the last inclusion. The study methods have been published previously [16]. In brief, educational hepatitis B sessions with the possibility for immediate HBV screening were scheduled after the regular meetings of Turkish organizations (e.g. Islamic mosque) located in Middle Limburg. Turkish migrants could also contact the culturally targeted, multilingual (Dutch-, English- and Turkish-speaking) first author for home visits and/or for an appointment at the hepatology outpatient clinic to sign an informed consent and be screened for HBV infection. The free outreach hepatitis B testing campaign was combined with risk factor assessment by a face-to-face questionnaire. Blood samples were tested for HBsAg, antibodies against HBsAg (anti-HBs) and hepatitis B core antibodies (anti-HBc) with an electrochemiluminescence assay (Cobas 8000 e602, Roche, Germany). Both participants and their general practitioner received the test result by letter and, supplementary to this, the general practitioners were advised to contact the HBsAg positive participants by phone. All patients who tested positive for HBsAg were referred to a hepatologist for a clinical work-up and treatment program according to the European clinical practice guidelines [17]. HBsAg positive patients were divided in four phases, taking into account the presence of hepatitis B e antigen (HBeAg), serum HBV DNA levels and serum alanine aminotransferase (ALT) values [17]. Persons susceptible to HBV infection were asked to consult their general practitioner for HBV vaccination.

### Questionnaire

Before hepatitis B testing, HBV risk factors were assessed through a face-to-face questionnaire (S1–S6 Figs). The questionnaire was available in the Dutch or Turkish language, and covered a total of 23 and 21 questions for males and females, respectively. The questionnaire did not cover information on intravenous drug use or sexual behaviour as these questions are sensitive and therefore socially censured; pressure to conform to societal norms could cause self-reports to be fraught with bias.

### Statistical analysis

The sample size calculation was performed with the aid of Epi Info® (version 7.2). Since data on country of birth in Middle Limburg are unavailable and data on nationality do not represent country of birth, the number of study participants per age group was calculated so that the results of the study group agree with that of the Turkish population living in Middle Limburg with a significance level of 5%. This is important for the extrapolation of the study results to the general Turkish population (Table 1).

To assess the impact of universal HBV vaccination in Belgium among Turkish SGM, we stratified HBsAg, anti-HBc and/or anti-HBs prevalence into two birth cohorts (1930–1986 and 1987–2001). This cutoff was chosen as the free-of-charge HBV vaccination program in infants since September 1999 with catch-up vaccination for 10–13 year-olds covered children born after 1987 in Belgium [18].

**Table 1. Composition of the study group compared with the composition of the Turkish population living in Middle Limburg regarding age and gender.**

| Age (years) | Total | 18–39 | 40–59 | $\geq$ 60 | Male | Female |
|---|---|---|---|---|---|---|
| Study group (N) | 1,081 | 433 | 511 | 137 | 468 | 613 |
| Study group (%) | 100 | 40.1 | 47.3 | 12.6 | 43.3 | 56.7 |
| Turkish Middle Limburg population (%) | 100 | 53.9 | 35.4 | 10.7 | 50.5 | 49.5 |

Since participants could be part of the same household, resulting in violation of the assumption of independent observations for classical tests such as logistic regression, univariate generalized estimating equations (GEE) were used to investigate the influence of the different variables on the presence of past or current HBV infection (anti-HBc positive), chronic HBV infection (HBsAg positive) and vaccination status (solely anti-HBs positive). The correlation structure in these GEE is assumed to be of the 'compound symmetry' type, specifying that all observations within the same cluster are equally correlated. Although this might not be the correct assumption, GEE has been shown to be robust to misspecification of the working correlation structure [19].

Variables showing significant association ($p < .20$) in the univariate analyses were subsequently included in a multiple GEE model. As origin and educational level of mother and father were highly correlated, only maternal factors were included in the models. Backward selection was used to obtain a final model. In order to correct for differences between the sample and the Middle-Limburg population, weights were included in all GEE models. Weighting was done based on the combination of age and gender.

## Ethical approval

The study was undertaken in accordance with Good Clinical Practice guidelines and the Declaration of Helsinki. The protocol was approved by the medical ethics committees of Hasselt University and Hospital East-Limburg (17-039U). All participants gave their informed consent in writing prior to inclusion.

## Results

In total 1,081 Turkish migrants were tested for HBV infection; 1,047 of the 1,079 (97.0%) persons present at the educational meetings in a Turkish organization and all 34 (100.0%) participants present at home visits did go for testing. Less than half of the study participants were male (43.3%), and the mean age was 44 ± 13.7 years. The majority of the participants were born in Turkey (58.1%), 41.9% were born in Belgium.

Blood samples were insufficient for anti-HBs testing in four participants. In 21.5% (232/1,081) there was evidence of a past or recent HBV infection (anti-HBc positive), 2.4% (26/1,081) had a chronic HBV infection (HBsAg positive), 22.9% (247/1,077) of the samples were solely anti-HBs positive ('vaccinated' serostatus) and 55.5% (598/1,077) appeared to be susceptible for HBV infection (negative HBV markers).

### Past or recent HBV infection

Past or recent HBV infection in FGM was apparent in 206/628 (32.8%) with differences between males and females, i.e. 40.4% (110/272) vs 27.0% (96/356), respectively, $p = .001$. Among FGM, anti-HBc prevalence in 18–39 year-olds, 40–59 year-olds and >60 year-olds was 13.2% (14/106), 31.9% (123/385) and 50.4% (69/137), respectively, $p < .001$.

Gender difference regarding anti-HBc prevalence was not seen in Turkish SGM, i.e. 5.6% (11/196) in males vs 5.8% (15/257) in females, $p = 1.000$. Anti-HBc prevalence in Turkish

SGM born after 1987 was 1.1% (2/190), and was higher for those SGM born before 1987, i.e. 9.1% (24/263), $p < .001$. Older age was associated with past or recent HBV infection in SGM, i.e. 4.0% (13/327) in 18–39 year-olds vs 10.3% (13/126) in 40–59 year-olds, $p = .018$. There were no SGM 60 years or older.

Factors associated with past or recent HBV infection in the weighted univariate GEE are shown in Table 2. From the weighted multiple GEE analyses as shown in Table 3, it can be seen that male gender ($p = .021$), older age ($p < .001$), FGM ($p < .001$), lower (i.e. none or primary) educational level of the mother ($p = .003$), HBV infected mother ($p = .008$), HBV infected siblings ($p = .002$), HBV infected other family member ($p = .004$), gynaecological examination in Turkey or unsafe male circumcision ($p = .032$), dental treatment in Turkey ($p = .049$), and treatment with needles in Turkey ($p = .012$) significantly increased the odds of anti-HBc positivity.

Multiple GEE analyses were also conducted for males and females separately as well as for FGM and SGM (S1–S4 Tables). The final model for males and females included age ($p = .006$ and $p < .001$), FGM ($p < .001$ and $p < .001$) and treatment with needles in Turkey ($p = .030$ and $p = .003$). In males, HBV infected siblings ($p = .022$) and circumcision not conducted by a medical doctor were additional independent risk factors.

Independent risk factors for recent or past HBV infection among FGM included age ($p < .001$), HBV infected mother ($p = .042$), HBV infected siblings ($p = .024$), gynaecological examination in Turkey or unsafe male circumcision ($p < .001$), and treatment with needles in Turkey ($p = .003$). Among SGM, risk factors included HBV infected other family member ($p = .010$) and treatment with needles in Turkey ($p = .030$).

Sixty-five patients with past or recent HBV infection had a family member with HBV infection: among seven (10.8%) patients only the partner was infected, 14 (21.5%) with HBV infected mother only, one (1.5%) with HBV infected father only, 16 (24.6%) with HBV infected siblings only, 19 (29.2%) had another family member with HBV infection, three (4.6%) with HBV infection among mother and siblings, one (1.5%) with HBV infected father and siblings, and four (6.2%) with HBV infected siblings and other family member. Therefore HBV infected mother could potentially explain 17/65 (26.2%) anti-HBc positive cases with a positive family history of HBV infection.

## Chronic HBV infection

Of the 26 HBsAg positive individuals, 19 were FGM, leading to a HBsAg prevalence of 3.0% (19/628) in FGM. Compared to females, male FGM had a higher prevalence of chronic HBV infection (5.5% (15/272) in males vs 1.1% (4/356) in females, $p = .002$).

Gender difference regarding chronic HBV infection was not seen in Turkish SGM, i.e. 1.5% (3/196) in males vs 1.6% (4/257) in females, $p = 1.000$. Overall, 1.5% (7/453) SGM were HBsAg positive; only one out of seven HBsAg positive SGM was born after 1987. This corresponds to a HBsAg positivity of 0.5% (1/190) and 2.3% (6/263) in those born after and before 1987, respectively, $p = .135$. All but one had HBV infection in the family: one with HBV infection in mother only, two with HBV infection in siblings only, one with HBV infection in another family member only and two with HBV infection in siblings and another family member.

S5 Table shows factors associated with chronic HBV infection in the weighted univariate GEE. From the weighted multiple GEE analyses as shown in Table 4, it can be seen that older age ($p = .022$) and gynaecological examination in Turkey or unsafe male circumcision ($p = .004$) significantly increased the odds, while body piercing/tattooing/earlobe perforation in Turkey resulted in a lower odds of chronic HBV infection ($p = .009$).

**Table 2. Prevalence of past or recent hepatitis B virus infection by different risk factors among the total study population (n = 1,081) (weighted univariate GEE).**

| | n | N | Prevalence (%) | P value | Crude OR (95% CI) |
|---|---|---|---|---|---|
| Overall | 232 | 1,081 | 21.5% | - | - |
| Gender | | | | .038 | |
| Male | 121 | 468 | 25.9% | | 1.38 (1.02–1.86) |
| Female | 111 | 613 | 18.1% | | (ref) |
| Age group | | | | < .001 | |
| 18–39 years | 27 | 433 | 6.2% | | (ref) |
| 40–59 years | 136 | 511 | 26.6% | | 5.73 (3.67–8.92) |
| ≥ 60 years | 69 | 137 | 50.4% | | 16.26 (9.55–27.67) |
| Ethnicity | | | | < .001 | |
| FGM | 206 | 628 | 32.8% | | 8.84 (5.59–13.97) |
| SGM | 26 | 453 | 5.7% | | (ref) |
| Year of immigration (if FGM) | | | | < .001 | |
| Before 1987 | 144 | 363 | 39.7% | | 2.33 (1.58–3.42) |
| Year 1987 or later | 61 | 264 | 23.1% | | (ref) |
| Southeastern Anatolian origin of mother | | | | .470 | |
| Yes | 15 | 59 | 25.4% | | 1.28 (0.68–2.42) |
| No | 216 | 971 | 22.2% | | (ref) |
| Mother's educational level | | | | < .001 | |
| None | 167 | 570 | 29.3% | | 7.65 (0.96–60.73) |
| Primary school | 61 | 380 | 16.5% | | 3.57 (0.45–28.64) |
| Secondary school | 3 | 123 | 2.4% | | 0.37 (0.03–4.15) |
| High school/University | 1 | 13 | 7.7% | | (ref) |
| HBV infected partner | | | | .698 | |
| Yes | 7 | 25 | 28.0% | | 1.23 (0.43–3.50) |
| No/Unknown | 225 | 1,056 | 21.3% | | (ref) |
| HBV infected mother | | | | .005 | |
| Yes | 17 | 37 | 45.9% | | 4.21 (1.98–8.99) |
| No/Unknown | 215 | 1,044 | 20.6% | | (ref) |
| HBV infected father | | | | .016 | |
| Yes | 2 | 24 | 8.3% | | 0.24 (0.05–1.17) |
| No/Unknown | 230 | 1,057 | 21.8% | | (ref) |
| HBV infected siblings | | | | .003 | |
| Yes | 21 | 45 | 46.7% | | 3.77 (1.81–7.87) |
| No/Unknown | 211 | 1,036 | 20.4% | | (ref) |
| HBV infected other family member | | | | .008 | |
| Yes | 26 | 70 | 37.1% | | 2.65 (1.47–4.80) |
| No/Unknown | 206 | 1,011 | 20.4% | | (ref) |
| Sharing toothbrushes regularly | | | | < .001 | |
| Yes | 62 | 188 | 33.0% | | 2.35 (1.59–3.47) |
| No | 170 | 892 | 19.1% | | (ref) |
| Sharing nail clippers | | | | .296 | |
| Yes | 208 | 991 | 21.0% | | 0.74 (0.43–1.25) |
| No | 24 | 89 | 27.0% | | (ref) |
| Sharing razors | | | | .609 | |
| Yes | 36 | 170 | 21.2% | | 1.13 (0.73–1.75) |
| No | 196 | 910 | 21.5% | | (ref) |
| Sharing used towels | | | | .164 | |

*(Continued)*

**Table 2.** (Continued)

| | n | N | Prevalence (%) | *P* value | Crude OR (95% CI) |
|---|---|---|---|---|---|
| Yes | 206 | 945 | 21.8% | | 1.38 (0.85–2.24) |
| No | 26 | 135 | 19.3% | | (ref) |
| Eating from the same plate | | | | .423 | |
| Yes | 201 | 927 | 21.7% | | 1.19 (0.77–1.84) |
| No | 31 | 153 | 20.3% | | (ref) |
| Gynaecological examination in Turkey (if female) | | | | .003 | |
| Yes | 30 | 99 | 30.3% | | 2.73 (1.60–4.67) |
| No | 81 | 514 | 15.8% | | (ref) |
| Circumcision (if male) | | | | .004 | |
| Collective | 44 | 118 | 37.3% | | 2.18 (1.34–3.56) |
| Alone | 66 | 305 | 21.6% | | (ref) |
| Circumcision not carried out by medical doctor (if male) | | | | < .001 | |
| Yes | 84 | 228 | 36.8% | | 5.24 (2.76–9.94) |
| No | 14 | 153 | 9.2% | | (ref) |
| Gynaecological examination in Turkey or unsafe male circumcision | | | | < .001 | |
| Yes | 115 | 330 | 34.9% | | 2.92 (2.12–4.02) |
| No | 117 | 751 | 15.6% | | (ref) |
| Blood transfusion | | | | .062 | |
| Yes | 44 | 167 | 26.3% | | 1.52 (1.01–2.27) |
| No | 188 | 914 | 20.6% | | (ref) |
| Dental treatment in Turkey | | | | < .001 | |
| Yes | 147 | 430 | 34.2% | | 4.36 (3.17–5.98) |
| No | 85 | 651 | 13.1% | | (ref) |
| Surgery in Turkey | | | | < .001 | |
| Yes | 50 | 133 | 37.6% | | 3.12 (2.05–4.76) |
| No | 182 | 948 | 19.2% | | (ref) |
| Treatment with needles in Turkey | | | | < .001 | |
| Yes | 107 | 322 | 33.2% | | 2.82 (2.04–3.91) |
| No | 125 | 759 | 16.5% | | (ref) |
| Body piercing/tattooing/earlobe perforation in Turkey | | | | .009 | |
| Yes | 84 | 327 | 25.7% | | 1.58 (1.14–2.20) |
| No | 148 | 754 | 19.6% | | (ref) |
| Fish spa treatment | | | | .058 | |
| Yes in Turkey | 9 | 64 | 14.1% | | 0.54 (0.25–1.14) |
| No/Other | 223 | 1,017 | 21.9% | | (ref) |

Abbreviation: GEE: generalized estimating equations; OR: odds ratio; CI: confidence interval; HBV: hepatitis B virus; FGM: first-generation migrants; SGM: second-generation migrants.

First-generation migrants: foreign-born individuals; second-generation migrants: individuals born in Belgium with foreign-born parents; unsafe male circumcision: collective circumcision and/or circumcision not carried out by medical doctor.

**Follow-up in patients with chronic HBV infection.** Eleven of 26 chronically infected patients were unaware of their HBV status and the other 15 were already linked to care (Fig 1). Thus, the percentage of newly diagnosed HBsAg positive patients was 1.02% (11/1,081). Two patients were referred to another hospital, one could not be linked to care due to incorrect contact details and the remaining eight patients were seen at the Department of Gastroenterology and Hepatology of Hospital East-Limburg.

**Table 3. Association of past or recent hepatitis B virus infection to different risk factors among the total study population (n = 1,081) (weighted multiple GEE model).**

| Parameter | | Estimate (SE) | p-value | aOR (95% CI) |
|---|---|---|---|---|
| (intercept) | | -4.39 (0.79) | | |
| Gender | | | .021 | |
| | Male (vs Female) | 0.46 (0.19) | | 1.58 (1.08–2.31) |
| Age group | | | < .001 | |
| | 40–59 years (vs 18–39 years) | 0.80 (0.25) | | 2.21 (1.36–3.62) |
| | ≥ 60 years (vs 18–39 years) | 1.50 (0.32) | | 4.50 (2.39–8.47) |
| Ethnicity | | | < .001 | |
| | FGM (vs SGM) | 1.22 (0.26) | | 3.40 (2.03–5.69) |
| Mother's educational level | | | .003 | |
| | None (vs High school/University) | 0.61 (0.80) | | 1.84 (0.38–8.88) |
| | Primary (vs High school/University) | 0.53 (0.80) | | 1.70 (0.35–8.20) |
| | Secondary (vs High school/University) | -1.18 (1.05) | | 0.31 (0.04–2.41) |
| HBV infected mother | | | .008 | |
| | Yes (vs No/Unknown) | 1.51 (0.45) | | 4.52 (1.89–10.84) |
| HBV infected siblings | | | .002 | |
| | Yes (vs No/Unknown) | 1.53 (0.40) | | 4.61 (2.13–10.01) |
| HBV infected other family member | | | .004 | |
| | Yes (vs No/Unknown) | 1.35 (0.39) | | 3.86 (1.80–8.30) |
| Gynaecological examination in Turkey or unsafe male circumcision | | | .032 | |
| | Yes (vs No) | 0.41 (0.19) | | 1.51 (1.04–2.21) |
| Dental treatment in Turkey | | | .049 | |
| | Yes (vs No) | 0.40 (0.20) | | 1.50 (1.01–2.21) |
| Treatment with needles in Turkey | | | .012 | |
| | Yes (vs No) | 0.49 (0.19) | | 1.63 (1.13–2.36) |

Abbreviations: GEE: generalized estimating equations; SE: standard error; aOR: adjusted odds ratio; CI: confidence interval; HBV: hepatitis B virus; FGM: first-generation migrants; SGM: second-generation migrants.

First-generation migrants: foreign-born individuals; second-generation migrants: individuals born in Belgium with foreign-born parents; unsafe male circumcision: collective circumcision and/or circumcision not carried out by medical doctor.

**Table 4. Association of chronic HBV infection to different risk factors among the total study population (n = 1,081) (weighted multiple GEE).**

| Parameter | | Estimate (SE) | p-value | aOR (95% CI) |
|---|---|---|---|---|
| (intercept) | | -5.28 (0.65) | | |
| Age group | | | .022 | |
| | 40–59 years (vs 18–39 years) | 1.28 (0.64) | | 3.59 (1.03–12.48) |
| | ≥ 60 years (vs 18–39 years) | 1.49 (0.81) | | 4.45 (0.90–21.92) |
| Gynaecological examination in Turkey or unsafe male circumcision | | | .004 | |
| | Yes (vs No) | 1.48 (0.46) | | 4.37 (1.78–10.71) |
| Body piercing/tattooing/earlobe perforation in Turkey | | | .009 | |
| | Yes (vs No) | -1.13 (0.57) | | 0.32 (0.11–0.99) |

Abbreviations: GEE: generalized estimating equations; SE: standard error; aOR: adjusted odds ratio; CI: confidence interval.

Unsafe male circumcision: collective circumcision and/or circumcision not carried out by medical doctor.

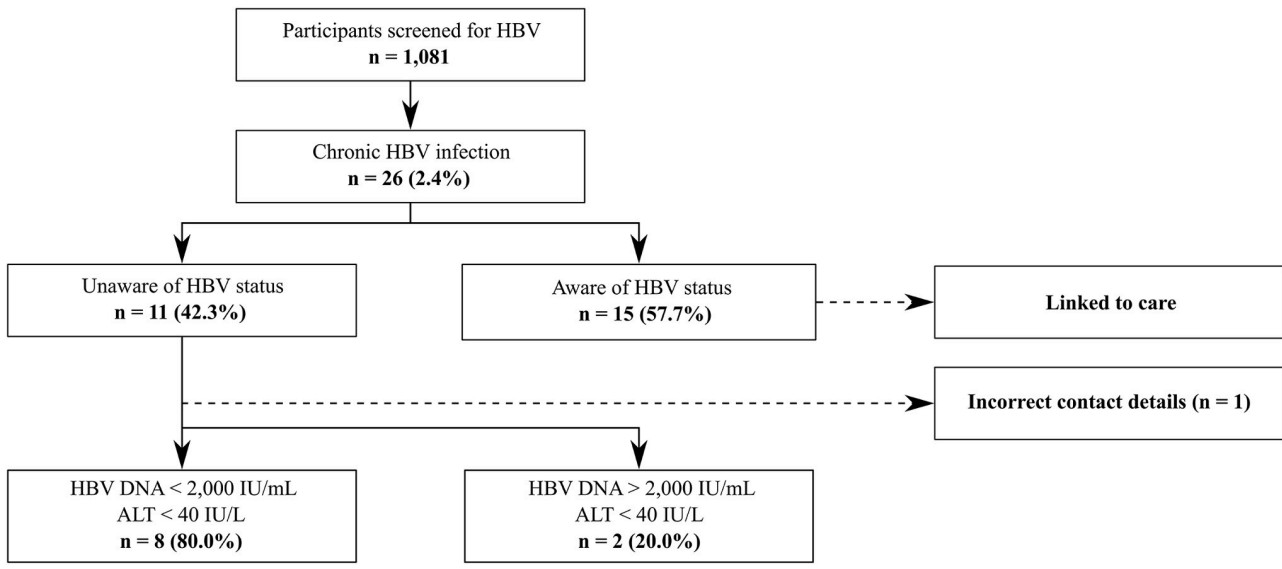

**Fig 1. Follow-up results in patients with chronic HBV infection.** Abbreviations: HBV: hepatitis B virus; ALT: alanine aminotransferase.

Out of the 15 patients that were already linked to care, 11 were classified as HBeAg-negative chronic infection, three as HBeAg-negative chronic hepatitis and one with HBeAg-positive chronic hepatitis. Both patients referred to another hospital and the eight patients evaluated in our hospital were categorized into two groups according to the HBV DNA ($\geq$ 2,000 IU/mL vs < 2,000 IU/mL) and ALT level (> 40 IU/L vs $\leq$ 40 IU/L) at the first visit (Fig 1). There was no evidence of cirrhosis or HCC on abdominal ultrasound. Advanced liver disease was suspected in one patient based on transient elastography value $\geq$ 10 kPa. However, liver biopsy revealed F1 (portal fibrosis without septa) and the presence of nonalcoholic steatohepatitis (NASH). Within one year follow-up, none of the ten newly HBsAg positive patients had an indication for antiviral treatment. Two patients were lost to follow-up after the first visit for reasons unknown.

## Solely anti-HBs positive

Regarding our question on HBV vaccination, a total of 22.2% (239/1,076) indicated to be vaccinated, 26.9% (289/1,076) were not vaccinated and history of HBV vaccination was unknown in the remaining 50.9% (548/1,076) participants. Among those with a history of vaccination (n = 239), 29 (12.1%) and 1 (0.4%) were anti-HBc positive and HBsAg positive, respectively.

Solely anti-HBs positivity in Turkish SGM born after 1987 was 72.1% (137/190), and was lower for those SGM born before 1987 (21.5%, 56/261), $p$ < .001. S6 Table shows factors associated with vaccination status in the weighted univariate GEE. For solely anti-HBs positivity (i.e. vaccination status), the final model included gender ($p$ = .008), age group ($p$ < .001), ethnicity ($p$ < .001), mother's educational level ($p$ = .007) and vaccination history ($p$ < .001) (Table 5).

## Susceptible for HBV infection

Six months after the last inclusion, 19.1% (44/230 susceptible individuals with follow-up information on vaccination) of the participants susceptible for HBV infection initiated vaccination.

**Table 5. Association of vaccination status (solely anti-HBs positive) to different risk factors among the total study population with information on anti-HBs (n = 1,077) (weighted multiple GEE).**

| Parameter | | Estimate (SE) | p-value | aOR (95% CI) |
|---|---|---|---|---|
| (intercept) | | 1.50 (0.69) | | |
| Gender | | | .008 | |
| | Male (vs Female) | -0.54 (0.20) | | 0.58 (0.39–0.86) |
| Age group | | | < .001 | |
| | 40–59 years (vs 18–39 years) | -1.30 (0.23) | | 0.27 (0.18–0.43) |
| | ≥ 60 years (vs 18–39 years) | -1.74 (0.54) | | 0.18 (0.06–0.51) |
| Ethnicity | | | < .001 | |
| | FGM (vs SGM) | -1.18 (0.22) | | 0.31 (0.20–0.47) |
| Mother's educational level | | | .007 | |
| | None (vs High school/University) | -1.51 (0.68) | | 0.22 (0.06–0.83) |
| | Primary (vs High school/University) | -1.23 (0.68) | | 0.29 (0.08–1.11) |
| | Secondary (vs High school/University) | -0.61 (0.69) | | 0.54 (0.14–2.11) |
| Vaccination history | | | < .001 | |
| | Not vaccinated/Unknown (vs Vaccinated) | -1.90 (0.21) | | 0.15 (0.10–0.22) |

Abbreviations: GEE: generalized estimating equations; SE: standard error; aOR: adjusted odds ratio; CI: confidence interval; FGM: first-generation migrants; SGM: second-generation migrants.

First-generation migrants: foreign-born individuals; second-generation migrants: individuals born in Belgium with foreign-born parents; unsafe male circumcision: collective circumcision and/or circumcision not carried out by medical doctor.

## Discussion

In addition to the screening of FGM, i.e. those born in Turkey, our study is one of the first hepatitis B screening projects in Europe targeting a substantial proportion of SGM, i.e. those born in Belgium with a foreign-born parent. First, we show that outreach testing was well-accepted in Turkish migrants. Although Turkish migrants could opt for screening during home visits or at the hepatology outpatient clinic, the majority (97%) were tested immediately after educational sessions in Islamic mosques and Turkish organizations.

Our finding of 3% HBsAg positivity in Turkish FGM is in line with previous findings reported in the Netherlands and Germany (HBsAg prevalence varying from 3–5%), and reflects the high prevalence in the country of origin [10, 12, 13, 20]. However, the 1.5% prevalence of chronic HBV infection in Turkish SGM is higher than the reported 0.7% HBsAg prevalence in native Belgians [6–8]. This finding is novel and might in part be explained by the interaction of high prevalence in the country of origin and suboptimal coverage by primary prevention strategies in the country of birth. None of the seven cases of SGM with chronic HBV infection in our study were born after 1999, which is the year universal HBV vaccination in infants was implemented in Belgium. However, although six HBsAg positive SGM were born before 1987, one was born between 1987 and 1999, a birth cohort that should have been covered by the catch-up vaccination in 10–13 year-olds since 1999. In this matter, lowest immunization coverage rates in Belgium were found in those born between 1990 and 1998 [21]. Moreover, prior studies indicated that non-European origin and low educational level of the mother were associated with a decreased odds of vaccination coverage in children [22, 23]. In our study, mother's educational level was associated with vaccination coverage.

In our Turkish migrant study population, HBV infection in the mother could potentially explain less than one third of the patients with past or recent HBV infection. HBV infection in siblings possibly accounted for another major part of past or recent HBV infection in our

study. In Turkey, perinatal transmission of HBV infection is uncommon as HBeAg positivity in pregnant chronic HBV patients is very low [24]. The infection is mostly acquired in childhood through horizontal transmission, i.e. that occurring through non-sexual and non-parenteral contact beyond six months of age, in childhood and pre-adolescent period [15, 24]. In the Turkish families, most of the mothers are housewives and mostly stay at home as caretakers for the children. Although the mother is important for horizontal HBV transmission through non-sexual close contact, sibling-to-sibling horizontal transmission might also be an important route of HBV infection [25–27]. Siblings share the same environment, experience the same cultural and family traditions, which may be associated with increased risks of transmission.

Physicians from countries with a selective vaccination program targeting groups at risk of HBV infection only, instead of adding a universal vaccination program, should be aware of horizontal transmission through non-sexual close contact [28]. Children are more likely to have contact with each other's body fluid and are therefore at risk of horizontal transmission through non-sexual close contact. Skin lesions or sharing contaminated material such as towels, toothbrushes or razor blades has been shown to play a role in horizontal transmission through non-sexual close contact [24]. In this matter, sharing toothbrushes regularly was univariately associated with past or recent HBV infection in our study population. However, this association diminished when accounting for other risk factors. The most important independent risk factors for past or recent HBV infection found in our study were male gender, older age, being a FGM, low educational level of the mother, a family history of HBV infection, gynaecological examination in Turkey or unsafe male circumcision, dental treatment in Turkey and treatment with needles in Turkey. These findings are in line with previous studies conducted in Turkish migrants living in Europe and studies conducted in Turkey [12, 13, 20, 24, 25].

For a screening program aimed at secondary prevention of HBV infection, access to care is inevitable to reduce the economic burden of HBV-related complications. In Greece, it is estimated that less than half of the patients diagnosed with HBV were referred and received the appropriate care [29]. This number was between 40% and 66% in the USA [30]. Most screening projects provide data on the HBsAg prevalence but do not assess referral, disease stage and initiation of antiviral therapy. We showed a successful referral of more than 90% newly diagnosed patients with chronic HBV infection. The implementation of a pre-screening phase with awareness activities among general practitioners and delegates of Turkish organizations as well as the impact of educational meetings on the Turkish migrant population to make an informed decision to participate in screening, might explain the good compliance to referral [11]. Within one year of follow-up, all newly diagnosed patients had HBeAg-negative chronic infection based on normal ALT levels, low HBV DNA levels, HBeAg negative status and no significant fibrosis [31]. During follow-up, none had an indication for antiviral therapy, but continued monitoring was still recommended for risk of HBV reactivation, advanced liver disease and HCC, especially those with HBV DNA levels $> 2,000$ IU/mL [31–33].

Less than one in five susceptible individuals initiated hepatitis B vaccination. This is lower than the reported 52–89% among migrant populations in the USA and might be explained by 1) dissimilar populations, 2) free of charge vaccination or vaccination at a reduced price and 3) the use of a combination of letters, phone calls, e-mail and in-person appointments to contact susceptible individuals for vaccination in these studies [11, 34–38]. This is in contrast to our study, in which susceptible individuals were invited by a letter to consult their general practitioner for hepatitis B vaccination in accordance with standard practices in Belgium. The uptake of vaccination was also low (22%) in an outreach screening project among Chinese migrants in the Netherlands [39].

This study had a few limitations. First, this study could be underpowered to find significant associations between certain risk factors and hepatitis B viral markers, as sample size calculation was performed in such a way that the distribution per age group was similar to that of the Turkish population in Middle Limburg. Second, although a face-to-face questionnaire could limit the risk of misinterpretation and incomplete entries, certain socially undesirable behaviours could have been underreported. In this matter, the questionnaire did not cover questions regarding intravenous drug use, men who have sex with men and multiple unsafe heterosexual contacts since the answers to these questions are ought to be unreliable in this Turkish population in which the majority of people identify with Islam. Third, this study describes the results of hepatitis B screening and risk factor assessment in the Turkish population in Middle Limburg, Belgium. It is unknown whether observations from our study could be extrapolated for the European region. However, the collection of several sociodemographic factors and HBV risk factors in the current study allow a detailed characterization of the study population and consequently in-depth comparisons with other studies could be established.

In conclusion, we found that outreach testing and referral to specialist care was clinically effective in the Turkish migrant population. We confirmed that HBV prevalence among Turkish FGM reflects the high prevalence in the country of origin. An important finding was the relatively high infection prevalence in Turkish SGM compared with prevalence estimates for the general population in Belgium and other European countries. Given WHO's ambitious goal to eliminate viral hepatitis by 2030, national HBV screening for Turkish FGM should be implemented and might be considered in SGM not covered by primary prevention strategies, taking into account the asymptomatic nature of disease and misleading normal levels of ALT. Screening projects aimed at SGM are needed to further inform future screening policies for this population.

## Supporting information

**S1 Fig. Questionnaire for males (Dutch version).**
(PDF)

**S2 Fig. Questionnaire for males (Turkish version).**
(PDF)

**S3 Fig. Questionnaire for males (English version).**
(PDF)

**S4 Fig. Questionnaire for females (Dutch version).**
(PDF)

**S5 Fig. Questionnaire for females (Turkish version).**
(PDF)

**S6 Fig. Questionnaire for females (English version).**
(PDF)

**S1 Table. Association of past or recent hepatitis B virus infection to different risk factors among the male study population (n = 468) (weighted GEE model).**
(PDF)

**S2 Table. Association of past or recent hepatitis B virus infection to different risk factors among the female study population (n = 623) (weighted GEE model).**
(PDF)

**S3 Table. Association of past or recent hepatitis B virus infection to different risk factors among first-generation migrants (n = 628) (weighted GEE model).**
(PDF)

**S4 Table. Association of past or recent hepatitis B virus infection to different risk factors among second-generation migrants (n = 453) (weighted GEE model).**
(PDF)

**S5 Table. Prevalence of chronic hepatitis B virus infection by different risk factors among the total population (n = 1,081) (weighted univariate GEE).**
(PDF)

**S6 Table. Vaccination status (solely anti-HBs positive) by different risk factors among the total population with information on anti-HBs (n = 1,077) (weighted univariate GEE).**
(PDF)

## Acknowledgments

The authors would like to acknowledge the many community leaders, Imams and volunteers for their assistance in the campaign events. This study is part of the Limburg Clinical Research Center (LCRC) UHasselt-ZOL-Jessa, supported by the foundation Limburg Sterk Merk, province of Limburg, Flemish Government, Hasselt University, Jessa Hospital and Ziekenhuis Oost-Limburg. The research was performed within the School of Nutrition and Translational Research in Metabolism (NUTRIM), Maastricht University.

## Author Contributions

**Conceptualization:** Özgür M. Koc, Cécile Kremer, Niel Hens, Rob Bielen, Dana Busschots, Pierre Van Damme, Geert Robaeys.

**Data curation:** Özgür M. Koc, Geert Robaeys.

**Formal analysis:** Cécile Kremer, Niel Hens.

**Funding acquisition:** Özgür M. Koc, Geert Robaeys.

**Investigation:** Özgür M. Koc, Cécile Kremer, Niel Hens.

**Methodology:** Özgür M. Koc, Cécile Kremer, Niel Hens, Geert Robaeys.

**Project administration:** Özgür M. Koc, Geert Robaeys.

**Resources:** Özgür M. Koc, Geert Robaeys.

**Supervision:** Niel Hens, Pierre Van Damme, Geert Robaeys.

**Writing – original draft:** Özgür M. Koc.

**Writing – review & editing:** Cécile Kremer, Niel Hens, Rob Bielen, Dana Busschots, Pierre Van Damme, Geert Robaeys.

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
