## [Decision Letter · Decision Letter 0]

17 Mar 2020

PONE-D-20-00619

Early detection of chronic hepatitis B and risk factor assessment in Turkish migrants, Middle Limburg, Belgium.

PLOS ONE

Dear Dr. Koc,

Thank you for submitting your manuscript to PLOS ONE. After careful consideration, we feel that it has merit but does not fully meet PLOS ONE’s publication criteria as it currently stands. Therefore, we invite you to submit a revised version of the manuscript that addresses the points raised during the review process.

ACADEMIC EDITOR: 

The data is well presented and technically sounds. Both reviewers made good comments about the paper. Please read carefully the recommendation of reviewers and editor that is presented below. 

We would appreciate receiving your revised manuscript by May 01 2020 11:59PM. To enhance the reproducibility of your results, we recommend that if applicable you deposit your laboratory protocols in protocols.io, where a protocol can be assigned its own identifier (DOI) such that it can be cited independently in the future. For instructions see: http://journals.plos.org/plosone/s/submission-guidelines#loc-laboratory-protocols

We look forward to receiving your revised manuscript.

Kind regards,

Livia Melo Villar

Academic Editor

PLOS ONE

Journal Requirements:

2. Please provide additional details regarding participant consent. In the ethics statement in the Methods and online submission information, please ensure that you have specified (a) whether consent was informed and (b) what type you obtained (for instance, written or verbal, and if verbal, how it was documented and witnessed). If your study included minors, state whether you obtained consent from parents or guardians. If the need for consent was waived by the ethics committee, please include this information.

4. Please ensure you have thoroughly discussed any potential limitations of this study within the Discussion section.

6. Thank you for stating the following financial disclosure:

"This work was supported by Gilead Sciences by an unrestricted grant registered as V 2331 at Hasselt University."   

Please respond by return e-mail so that we can amend your financial disclosure and competing interests on your behalf.

7. Thank you for stating the following in the Competing Interests section:

"None"

We note that you received funding from a commercial source: "Gilead Sciences"

Additional Editor Comments (if provided):

Dear Author,

Thanks for sending the paper intitled: "Early detection of chronic hepatitis B and risk factor assessment in Turkish migrants, Middle Limburg, Belgium". This paper reports the prevalence of HBV infection in individuals with a Turkish background (FGM and SGM), the results of a questionnaire on risk factors, and the follow-up of patients referred to care. The data is well presented and technically sounds.

Each reviewer made comments to this revision that should be answered.

Specific Comments of Editor:

I also recommend to include more information regarding population recruitment in methods section.

Reviewers' comments:

Reviewer's Responses to Questions

**Comments to the Author**

1. Is the manuscript technically sound, and do the data support the conclusions?

Reviewer #1: Yes

Reviewer #2: Yes

2. Has the statistical analysis been performed appropriately and rigorously? 

Reviewer #1: I Don't Know

Reviewer #2: Yes

3. Have the authors made all data underlying the findings in their manuscript fully available?

Reviewer #1: Yes

Reviewer #2: Yes

4. Is the manuscript presented in an intelligible fashion and written in standard English?

Reviewer #1: Yes

Reviewer #2: Yes

5. Review Comments to the Author

Reviewer #1: The study is extremely relevant. Since immigrants are key populations in the elimination of HBV. These populations are often neglected and access to information and effective prevention and treatment is not provided to them. It is important to highlight the inclusion of those born in Belgium to immigrant parents. For, the customs, culture and prejudice against this population continue in this generation, which may favor a transmission network that needs to be interrupted in terms of global public health.

Study considerations:

I did not find the part of the title "Early detection of chronic hepatitis B" interesting.

Why do you think it was early detection? It would only be more correct, detection of chronic hepatitis B.

The summary does not speak of the most important risk factors found in the study.

In the introduction: line 92-94. Paragraph 3 is confused. Are you talking about all countries in general, or where did the study take place?

The questions reported in lines 137-139 are very important for the analysis of HBV risk factors, due to sexual and parenteral transmission. Despite being very particular factors, they need to be taken into account. They should be highlighted as limitations of the study and in an upcoming study to be evaluated.

This should be noted, that these factors were not considered to be risky, as the necessary information was not obtained.

Was the chronic infection reported on line 190 (HbsAg + and anti-HBc +)?

Reviewer #2: Dr Koc et al. examined the prevalence of HBV in Turkish migrants in Belgium and determined the risk factors.

The finding that the horizontal transmission seems an important route for the infection is intriguing.

The necessity for screening projects aimed at second-generation migrates is an important message.

The research is technically sound and the data are probably precious in this field.

I do not have any special comments on this study.

6. PLOS authors have the option to publish the peer review history of their article (what does this mean?). If published, this will include your full peer review and any attached files.

Reviewer #1: No

Reviewer #2: No

---

## [Author Response · Author response to Decision Letter 0]

21 Apr 2020

Answer: We changed accordingly. 

2. Please provide additional details regarding participant consent. In the ethics statement in the Methods and online submission information, please ensure that you have specified (a) whether consent was informed and (b) what type you obtained (for instance, written or verbal, and if verbal, how it was documented and witnessed). If your study included minors, state whether you obtained consent from parents or guardians. If the need for consent was waived by the ethics committee, please include this information.

Answer: Minors were not included in the current study as stated under Patients and methods. In the methods section we included the following sentence: All participants gave their informed consent in writing prior to inclusion. (see lines 210-211)

Answer: We used separate questionnaires for men and women, available in Dutch and Turkish language as previously stated in the manuscript. In addition to the original questionnaires we now provide the English version as supporting information (see S1 – S6 Figs).

4. Please ensure you have thoroughly discussed any potential limitations of this study within the Discussion section.

Answer: Study limitations are now depicted within the discussion section as follows: This study had a few limitations. First, this study could be underpowered to find significant associations between certain risk factors and hepatitis B viral markers, as sample size calculation was performed in such a way that the distribution per age group was similar to that of the Turkish population in Middle Limburg. Second, although a face-to-face questionnaire could limit the risk of misinterpretation and incomplete entries, certain socially undesirable behaviours could have been underreported. In this matter, the questionnaire did not cover questions regarding intravenous drug use, men who have sex with men and multiple unsafe heterosexual contacts since the answers to these questions are ought to be unreliable in this Turkish population in which the majority of people identify with Islam. Third, this study describes the results of hepatitis B screening and risk factor assessment in the Turkish population in Middle Limburg, Belgium. It is unknown whether observations from our study could be extrapolated for the European region. However, the collection of several sociodemographic factors and HBV risk factors in the current study allow a detailed characterization of the study population and consequently in-depth comparisons with other studies could be established. (see lines 425-439)

Answer: Data will be made available in the following repository: Koc, ÖM et al. (2020), Early detection of chronic hepatitis B and risk factor assessment in Turkish migrants, Middle Limburg, Belgium., Dryad, Dataset, https://doi.org/10.5061/dryad.4f4qrfj87

6. Thank you for stating the following financial disclosure: "This work was supported by Gilead Sciences by an unrestricted grant registered as V 2331 at Hasselt University." 

Please state what role the funders took in the study. If the funders had no role, please state: "The funders had no role in study design, data collection and analysis, decision to publish, or preparation of the manuscript." Please respond by return e-mail so that we can amend your financial disclosure and competing interests on your behalf.

Answer: the funders had no role in the study design, data collection and analysis, decision to publish, or preparation of the manuscript.

7. Thank you for stating the following in the Competing Interests section:

"None"

We note that you received funding from a commercial source: "Gilead Sciences"

Answer: We apologize for the incomplete conflicts of interest statement. Hereby we provide a complete list of competing interests:

ÖK received travel grants from Gilead Sciences and his institution received grants from Gilead Sciences, AbbVie, MSD and CyTuVax B.V. NH is holder of the chair in evidence-based vaccinology supported though a gift by Pfizer. All outside the submitted work. RB has received travel grants from AbbVie, Gilead Sciences and MSD to attend scientific congresses and research grants from Gilead and MSD. DB has received travel grants from AbbVie and a research grant from Gilead. PVD acts as chief and principal investigator for vaccine trials conducted on behalf of the University of Antwerp, for which the University obtains research grants from vaccine manufacturers; speakers fees for presentations on vaccines are paid directly to an educational fund held by the University of Antwerp. PVD receives no personal remuneration for this work. GR has received research grants from AbbVie, MSD, Janssen Pharmaceuticals, and has acted as a consultant/advisor for AbbVie, MSD, Gilead Sciences and Bristol-Myers Squibb.

This does not alter our adherence to PLOS ONE policies on sharing data and materials.

Answer: We have included captions for our 11 supporting files at the end of our manuscript. The citations were changed in line with the PLOS ONE requirements.

Editor Comments:

Thanks for sending the paper intitled: "Early detection of chronic hepatitis B and risk factor assessment in Turkish migrants, Middle Limburg, Belgium". This paper reports the prevalence of HBV infection in individuals with a Turkish background (FGM and SGM), the results of a questionnaire on risk factors, and the follow-up of patients referred to care. The data is well presented and technically sounds.

Each reviewer made comments to this revision that should be answered.

1. I also recommend to include more information regarding population recruitment in methods section.

Answer: For the interpretation of the study, we included the following sentence: In brief, educational hepatitis B sessions with the possibility for immediate HBV screening were scheduled after the regular meetings of Turkish organizations (e.g. Islamic mosque) located in Middle Limburg. Turkish migrants could also contact the culturally targeted, multilingual (Dutch‐, English‐ and Turkish‐speaking) first author for home visits and/or for an appointment at the hepatology outpatient clinic to sign an informed consent and be screened for HBV infection. (see lines 146-151)

Reviewer#1:

The study is extremely relevant. Since immigrants are key populations in the elimination of HBV. These populations are often neglected and access to information and effective prevention and treatment is not provided to them. It is important to highlight the inclusion of those born in Belgium to immigrant parents. For, the customs, culture and prejudice against this population continue in this generation, which may favor a transmission network that needs to be interrupted in terms of global public health.

1. I did not find the part of the title "Early detection of chronic hepatitis B" interesting.

Why do you think it was early detection? It would only be more correct, detection of chronic hepatitis B.

Answer: Our hepatitis B screening targeted at the Turkish migrant population is a form of secondary prevention aimed to reduce the impact of HBV infection. This is done by detecting HBV infection as soon as possible to prevent disease progression. It is well known that a chronically infected hepatitis B patient has no or minimal symptoms until the development of serious late complications such as cirrhosis, decompensated cirrhosis and hepatocellular carcinoma. Since none of the newly diagnosed hepatitis B patients in the current study had cirrhosis at presentation, we would like to underline the importance of early detection.

2. The summary does not speak of the most important risk factors found in the study.

Answer: You are correct. We included the most important risk factors in the summary. (see lines 65-69)

3. In the introduction: line 92-94. Paragraph 3 is confused. Are you talking about all countries in general, or where did the study take place?

Answer: This is a general statement indicating that migrants from endemic regions (HBsAg positivity 2-7.99% for intermediate endemic regions and >8% for high endemicity) are an important risk group in regions with low endemicity for hepatitis B (HBsAg positivity <2%), such as Western Europe.

4. The questions reported in lines 137-139 are very important for the analysis of HBV risk factors, due to sexual and parenteral transmission. Despite being very particular factors, they need to be taken into account. They should be highlighted as limitations of the study and in an upcoming study to be evaluated.

This should be noted, that these factors were not considered to be risky, as the necessary information was not obtained.

Answer: We agree with the reviewer that information regarding intravenous drug use and high-risk sexual behavior is very important for the analysis of HBV risk factors. However, this information was not collected since the answers to these sensitive questions are probably unreliable in the Turkish migrant population in which the majority are Muslims. 

In accordance with the journal requirements we included this as a limitation within the discussion. Study limitations are now depicted within the discussion section as follows: Second, although a face-to-face questionnaire could limit the risk of misinterpretation and incomplete entries, certain socially undesirable behaviours could have been underreported. In this matter, the questionnaire did not cover questions regarding intravenous drug use, men who have sex with men and multiple unsafe heterosexual contacts since the answers to these questions are ought to be unreliable in this Turkish population in which the majority of people identify with Islam. (see lines 428-432)

5. Was the chronic infection reported on line 190 (HbsAg + and anti-HBc +)?

Answer: HBsAg positivity was considered as chronic HBV infection in the current study. All HBsAg patients were also anti-HBc positive. Considering the prospective follow-up of HBsAg positive individuals, we indeed confirmed that they were chronically infected with HBV infection.

Reviewer #2: 

Dr Koc et al. examined the prevalence of HBV in Turkish migrants in Belgium and determined the risk factors.

The finding that the horizontal transmission seems an important route for the infection is intriguing.

The necessity for screening projects aimed at second-generation migrates is an important message.

The research is technically sound and the data are probably precious in this field.

I do not have any special comments on this study.

Answer: we would like to thank the reviewer for her/his kind remarks.

---

## [Editor Report · Decision Letter 1]

2 Jun 2020

Early detection of chronic hepatitis B and risk factor assessment in Turkish migrants, Middle Limburg, Belgium.

PONE-D-20-00619R1

Dear Dr. Koc,

We are pleased to inform you that your manuscript has been judged scientifically suitable for publication and will be formally accepted for publication once it complies with all outstanding technical requirements.

With kind regards,

Livia Melo Villar

Academic Editor

PLOS ONE

Additional Editor Comments (optional):

Dear Author,

The suggestions and corrections were made as requested.

Best regards

Livia Villar
---

## [Editor Report · Acceptance letter]

16 Jul 2020

PONE-D-20-00619R1 

Early detection of chronic hepatitis B and risk factor assessment in Turkish migrants, Middle Limburg, Belgium. 

Dear Dr. Koc:

I'm pleased to inform you that your manuscript has been deemed suitable for publication in PLOS ONE. Congratulations! Your manuscript is now with our production department. 

Kind regards, 

on behalf of

Dr. Livia Melo Villar 

Academic Editor

PLOS ONE